# How do question-answer exchanges among generations matter for children's happiness?

Junichi Hirose [1,2]*

**1** Multidisciplinary Science Cluster, Collaborative Community Studies Unit, Kochi University, Kochi, Japan,
**2** Research Institute for Future Design, Kochi University of Technology, Kochi, Japan

* hirose-junichi@kochi-u.ac.jp

## Abstract

Intergenerational exchange plays an increasing role in realizing sustainable societies. Question-answer exchanges are the trigger for individuals to initiate some intergenerational relations, and the literature has established that inquisitiveness (curiosity about something and someone different) contributes to people's generativity and happiness. However, little is known about how children's inquisitiveness influences their generative concern and happiness. We claim that inquisitiveness is essential for children to enhance their happiness and hypothesize that those who receive a positive response from adults tend to be inquisitive and express the signs of generativity. To empirically examine the hypothesis, we have statistically characterized inquisitiveness in relation to adult-child interaction, generativity (offering care for people and the natural environment) and happiness, using the data from a survey of 511 Japanese children between 9 and 14 years and by applying the revised generativity concern scale (GCS-R). The results show that inquisitiveness correlates with generativity and happiness, primarily that a positive response by adults to children's inquiries promotes their inquisitiveness through adult-child interactions. Our analysis shows that children's inquisitiveness, encouraged by adults' positive responses, is more significant in happiness than the generativity concern during childhood. Overall, the results suggest that adults responding positively to children's questions is essential for promoting inquisitiveness and increasing happiness.

## Introduction

Curiosity is an essential element of individuals' creativity, maturity and open-mindedness [1–4]. Children's tendency to ask questions, driven by their curiosity, shall be an initial step in building human relations and learning. In the literature, such a tendency is conceptualized as "inquisitiveness" representing curiosity about something and someone different [5, 6]. For instance, Frazier et al. [7] examine adult-child conversational exchanges by focusing on young children's questions and responses and claims that such communications provide essential bases for children's future life, especially their growth through human interactions. Regarding the role of adults in responding to children's curiosity, "generativity" has been known as the characteristic that helps the next generation grow and lead to a sustainable society [8–11].

**Funding:** This work was supported by General Incorporated Association Happiness Promotion Council (R3-0817) of JH. The publication

procedure was partly supported by the research fund of JH from Kochi University. The funders were engaged in the administrative procedure but not involved in the study design, data collection and analysis, decision to publish, or the manuscript preparation.

**Competing interests:** The authors have declared that no competing interests exist.

Moreover, having excellent relationships with family, friends and others contributes to people's generativity and happiness [9, 12–15]. However, little is known about how inquisitiveness influences people's generativity and happiness in their childhood. Therefore, this research addresses the role and mechanism of children's inquisitiveness in the emergence of generativity and their happiness through intergenerational communication.

Wellbeing represents the outcome of a "good life," where people are assumed to act and behave as happiness seekers [16–19]. Maslow [20] is the psychologist who proposed a wellbeing theory, i.e., Maslow's hypothesis, based on psychological needs and gratification processes, and implies that material wealth increases people's satisfaction with the basic needs stage but that other factors, such as psychological factors, are essential to people's overall wellbeing. To examine this hypothesis, some researchers have developed and refined happiness measurements, such as the subjective happiness scale (SHS) and satisfaction with life scale (SWLS) (see, e.g., [21–23]). Veenhoven [24] and Diener and Diener [25] empirically examine the hypothesis with cross-country data utilizing happiness scales and conclude that wealth can explain variations in happiness across countries; however, there should be some other crucial predictors. Following these works, the literature has focused on how happiness is associated with various cultural, sociodemographic and personal factors other than income or wealth, including gender, marital status, educational record, self-esteem, optimism and human relations [26–33]. Overall, aging, income, human relationships and personality traits are established to be the essential characteristics of happiness [34–40].

Erikson [8] puts forth the idea of generativity, which he defines as the focus on fostering and directing the development of the next generation in the context of life-course theory. This concept can manifest through parenting but is not limited to it [9]. Generativity can be demonstrated through various actions and behaviors, such as a desire to positively impact future generations or imparting valuable knowledge and skills to young people [10, 41, 42]. Several scales have been developed to quantify generative activities, behaviors and concerns, such as the Loyola generativity scale (LGS), generative concern scale (GCS) and the generative behavior checklist (GBC) [9, 13, 43, 44]. Utilizing these scales, many studies have empirically examined the relationship between generativity and happiness [13, 45–49]. Overall, these studies show that generativity is a robust and consistent predictor of happiness, controlling for prosociality and some other key sociodemographic factors, such as age, gender and marital status in the analyses [12, 14, 50–57].

As a concept, inquisitiveness represents curiosity about something and someone different, and those with such inquisitiveness tend to start communications with others by asking questions [58–62]. After the development by Facione et al. [63], Hirayama and Kusumi [58] and Hogan and Hogan [64] of inquisitiveness as a subscale to measure the disposition of critical thinking, many studies have been undertaken to address the questions of how inquisitive persons learn from and engage with people regardless of their backgrounds, positions and roles, and of how such behaviors may lead to creative problem-solving [60, 62, 64–69]. Hirayama and Kusumi [58] have conducted questionnaire surveys among 426 Japanese university students and analyzed the effects of their attitudes of critical thinking on the process of reaching conclusions. They find inquisitiveness as essential for reaching a conclusion that is not bounded by individuals' beliefs. Hirose and Kotani [70] also demonstrate that inquisitiveness positively correlates with how people enhance their generative concern and happiness by conducting questionnaires with 400 Japanese. Overall, inquisitiveness is an engine of increasing the motivation and behaviors in some situations, triggering communications with others and interactions with unfamiliar environments [59, 71–73].

No previous works have addressed how children's happiness is characterized by inquisitiveness and generativity, while these concepts are known to be concerned with how people

build and keep good relationships with their family, friends and others. Inquisitiveness is considered an essential factor that triggers communication even in a child and contributes to building pleasant human relations. Therefore, even among children, inquisitiveness is hypothesized to be a crucial determinant of happiness and generativity. This study aims to empirically examine the cognitive, noncognitive and sociodemographic factors of inquisitiveness, generativity and happiness in children using a single analytical framework, as described in Fig 1. Thus, we conduct questionnaire surveys with 511 Japanese participants between 9 and 14 years of age to collect data, following previous studies that have analyzed the relationships between behaviors and happiness with cross-sectional data [74–78]. The study collects data to analyze the relationship between children's curiosity, adult responses to children's questions, children's attitudes toward the younger and nature and children's happiness using the following measures: inquisitiveness (curiosity about and acceptance of something and someone different and/or new); costs/benefits of the consultation (positive effects of reaction in an appropriate manner); the revised generative concern (offering care for people and the natural environment); and subjective happiness. With these data, this study addresses the following three research questions. (1) How does children's inquisitiveness, along with generativity, affect their happiness? (2) Does their inquisitiveness play a role in generativity? (3) How does adults' manner of responding affect children's inquisitiveness?

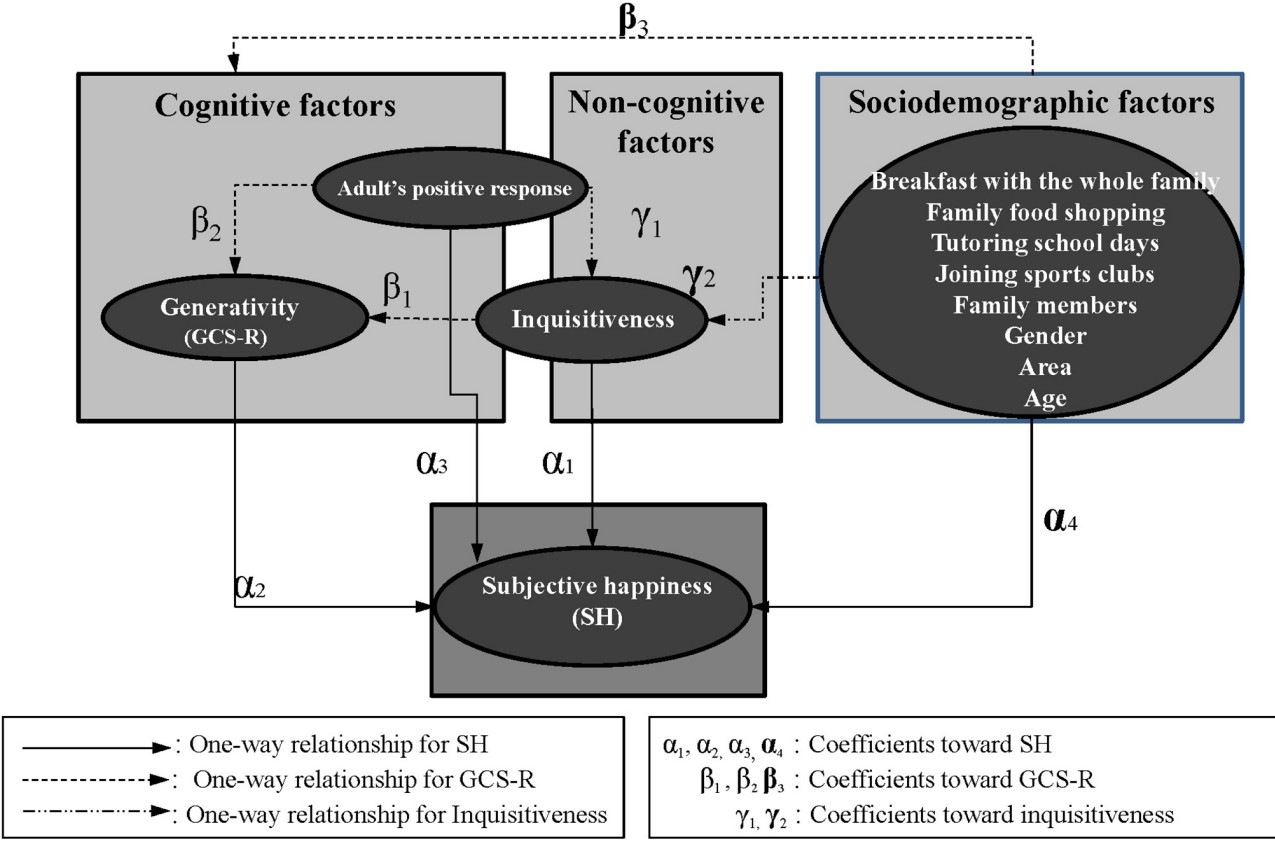

**Fig 1. A conceptual framework: A conceptual framework describing the relationships concerning subjective happiness (SH) among cognitive, noncognitive and sociodemographic factors.**

## Study regions

We conduct a questionnaire survey in Kochi, Japan. Kochi prefecture comprises the southwestern part of the island of Shikoku, facing the Pacific Ocean. Kochi prefecture has a population of 677, 888 (2022) and has a geographic area of 7,103km$^2$. Most of the region is mountainous, and in only a few areas, such as around Kochi and Nakamura, is there a coastal plain. City-A is the main commercial area, representing the urban areas of the prefecture with over 40% of its population. The population and total land area of city-A are 321, 910 and 309km$^2$, respectively. We focus on another two areas as non-urban areas: town-B and town-C. The population and total land areas of town-B (town-C) are 12, 000 (3, 200) and about 100km$^2$ (about 240km$^2$), respectively. We focus on another two areas as non-urban areas: town-B and town-C. The population and total land areas of town-B (town-C) are 12, 000 (3, 200) and about 100km$^2$ (about 240km$^2$), respectively. Previous studies have demonstrated that prosociality and wellbeing differ between urban and rural areas in Nepal and Bangladesh [49, 79, 80]. Therefore, we collect the samples from urban and non-urban areas, controlling for such possibilities in statistical analyses.

## Materials and methods

### Participants

The participants in the study are elementary and junior high school students living in a certain municipality in Kochi Prefecture, Shikoku, Japan. The participants' mean age is 12.32 years with the standard deviation = 14.23, ranging between 9 and 14 years. The percentage of female participants and their mean age are almost similar in urban and non-urban areas, 46% and 47%, as well as 12.61 years and 12.21 years, respectively. We have surveyed three regions (one categorized as urban and two non-urban) for the study because they possess different sociodemographic and geographical characteristics.

### Procedure

We recruited the participants by sending them invitation letters from the municipal education board offices. A total of 570 invitations were sent to schools in the three study sites. We collected 511 samples in city-A (140) and two rural areas of town-B (225) and town-C (146), respectively. After obtaining written consent for participation from the individual and his/her parents, we conducted a questionnaire survey in several school facilities in the study regions. The questionnaire used simple language that elementary school students could understand and was designed to be easy to answer on a touchscreen display. This survey collects a sample of 511 participants with information by asking them in the following order: (i) subjective happiness; (ii) inquisitiveness (curiosity about and acceptance of something and someone different and/or new); (iii) the revised generative concern (offering care for people and the natural environment); (iv) costs/benefits of the consultation (positive effects of reaction in an appropriate manner) and (v) sociodemographic factors, such as age, gender, attending tutoring school, joining sports clubs and household members. The variables we collected in this survey can be categorized into cognitive, non-cognitive and sociodemographic factors in relation to subjective happiness, as described in Fig 1. The data were collected between November and December 2021.

### Ethics statement

All procedures regarding this study were approved by the Kochi University Ethics Committee (No. 2021–43). Prior consent from the participants and their parents to participate in the

survey was provided through the Municipal Board of Education. The questionnaire was filled out once the participants were assured that no information collected in the survey would reveal their identity.

## Measures

**The subjective happiness scale (SHS).** We employ the subjective happiness scale (SHS) with a four-item measurement developed by Lyubomirsky and Lepper [23], where each item scores on a 7-point Likert scale, ranging from 1 = "Strongly disagree" to 7 = "Strongly agree," and the total scale scores are the sum of the four-item scores, ranging between 4 and 28. The SHS is one of the most commonly used measures of global happiness, in correlational studies including different self-report scales [81]. The first question is, "Generally, how do you consider yourself?" and its anchors are "not a very happy person" and "a very happy person." The second question is, "How do you consider yourself when you compare yourself to others?" and its anchors are "less happy" and "happier." The third and fourth questions correspond to a general description of a happy and/or unhappy person, where the participants make a choice to describe themselves best. In the items, "Some people are generally very happy. They enjoy life no matter what is going on, getting the most of everything. How much does this sentence describe you?" and "Some people are generally very happy. Although they are not depressed, they never seem as happy as they might be. How does this sentence describe you?"; the anchors are "not at all" and "a great deal," called unhappiness and general subjective happiness. The average of all items is the overall subjective happiness (OSH), while the fourth is reversely coded. We use OSH to represent "subjective wellbeing (SWB)" as happiness for regression analyses in this study.

**The revised generative concern scale (GCS-R).** Some generativity scales have been developed to measure individuals' differences considering their various aspects [14]. The Loyola generativity scale (LGS) is employed to assess "generative concern," as it is most commonly used in the literature (see, e.g., [9, 50, 52, 55, 82, 83]). Another popular scale for generativity is the generative behavior checklist (GBC) that scores on "generative behaviors" in the past two months [14, 43]. These two scales are established to display positive associations, demonstrating consistency between generative concern and behaviors [43]. With the Japanese cultural context in mind, Marushima and Arimitsu [84] have developed the revised generative concern scale (GCS-R) composed of three subscales; "creativity," "maintaining" and "offering." We have decided to apply the GCS-R, because we realize that some items in both the LGS and GBC scales shall be difficult for many Japanese children to answer because of their cultural difference and inexperience.

The revised generativity concern scale (GCS-R) is composed of three subscales: creativity, maintaining and offering [84]. We applied the offering subscale because questions in this subscale could help even children easily imagine a situation and answer them. The offering subscale is mainly concerned with the question about caring for people and their natural environment, and we have confirmed that this subscale is applicable for measuring the generative concern of extremely young individuals. The other two subscales (creativity and maintenance) are questions that focus on aspects of individuality and the transmission of memorable life experiences, respectively [84]. The items in the subscale included six statements, such as (1) "I try to offer a helping hand when I see someone in need," (2) "If anyone is sad, I want to cheer them up," (3) "I like taking care of people," (4) "I am willing to participate in volunteer activities," (5) "I properly listen to the other person's story," (6) "I take good care of younger people than me" and (7) "I am careful not to pollute the natural environment for the younger than me." The participants need to choose one of four options for each statement. "Zero,"

"one," "two" or "three" scores indicate how often the statement applies to the participants (Mark "zero" if a statement never applies, mark "three" if the statement applies very often or nearly always). The revised generativity concern score for each participant is computed as the sum of the scores for all seven items. The maximum possible range is between 0 and 28 and is calculated as the sum of the scores from the GCS-R questions.

**The inquisitiveness subscale.** We employ the inquisitiveness scale in this survey, which is a subscale developed by Hirayama and Kusumi [58] to measure the disposition of critical thinking. This instrument is used to assess one's disposition for curiosity about & acceptance of something and someone different and/or new [58, 85, 86]. This subscale consists of ten items, including (1) "I want to interact with people of various ways of thinking and learn a lot from them," (2) "I want to keep learning new things throughout my life," (3) "I like to challenge new things," (4) "I want to learn about various cultures," (5) "Learning how foreigners think is meaningful to me," (6) "I am interested in individuals who adopt different ways of thinking," (7) "I want to know more about any topic," (8) "I want to learn as much as possible, even if I do not know if it is useful," (9) "It is interesting to discuss with people who hold different ideas from what I do" and (10) "I want to ask someone if I do not know." The items are rated from 1 = "Strongly disagree" to 5 = "Strongly agree." The maximum possible range is between 10 and 50 and is calculated as the sum of the scores on the inquisitiveness questions. This subscale reliably measures the influence of people's behaviors and attitudes in many vital contexts [70, 85].

**The positive and negative effects subscale.** We employ "the positive and negative effects subscale" developed by [87] to measure the revised expected costs/benefits of the consultation. The concept of cost/benefit in consulting research has been introduced from the perspective of help-seeking behavior in social psychology [87, 88]. Nagai and Arai [87] have empirically examined the effect of the consultation behavior on 792 junior high school students in Japan. The positive aspect of this subscale consists of eight items, including (1) "When I ask some adults for advice, I can know how to solve my concerns," (2) "When I ask some adults for advice, they help me solve my concerns," (3) "When I ask some adults for advice, my concerns are solved," (4) "When I ask some adults for advice, I get good comments and advice from them," (5) "When I ask some adults for advice, my feelings are refreshed," (6) "When I ask some adults for advice, they answer my questions in good faith," (7) "When I ask some adults for advice, I would feel better," (8) "When I ask some adults for advice, I am encouraged by them." The items are rated from 1 = "Strongly disagree" to 5 = "Strongly agree.". The maximum possible range of the positive subscale is between 8 and 40.

The negative aspect of this subscale consists of six items, including (1) "When I ask some adults for advice, they say nasty things to me," (2) "When I ask some adults for advice, they make fun of me," (3) "When I ask some adults for advice, they do not take me seriously," (4) "When I ask some adults for advice, they easily dismiss the conversation," (5) "When I ask some adults for advice, they always disagree with my ideas" and (6) "When I ask some adults for advice, they always give deferent opinions." The items are rated from 1 = "Strongly disagree" to 5 = "Strongly agree." The maximum possible range of the negative subscale is between 6 and 30, and Cronbach's alpha for this measure is 0.90 in our sample. These subscales are reliable for influencing people's behaviors and attitudes in many vital contexts [87, 88].

## Data analysis

We use the cross-sectional data of the variables mentioned above as follows. Table 1 present the definitions of all variables used in the analysis. First, we characterize happiness in relation to generativity and inquisitiveness along with other factors. Second, we describe generativity

**Table 1. Variable definitions.**

| Variables | Descriptions |
|---|---|
| Gender | Gender is a dummy variable that takes 1 when the participant is a female; otherwise 0. |
| Age | Age is defined as years of age. |
| Household member | Household member is defined as the nember of the family. |
| Tutoring school days | This is defined as days of tutoring school per week (Range is between 0 and 7). |
| Joining sports clubs | This is a dummy variable that takes 1 when the participant joins sports clubs; otherwise 0. |
| Having breakfast with the whole family | This is defined as the number of days per week, on which the participants have breakfast with the whole family (Range is between 0 and 7). |
| Having dinner with whole family | This is defined as the number of days per week, on which the participants have dinner with the whole family (Range is between 0 and 7). |
| Family food shopping | This is defined as the number of days per week on which the participants go grocery shopping with family (Range is between 0 and 7). |
| Subjective happiness (SH) | SH is defined as the score on the overall subjective happiness (OSH) of the subjective happiness scale (Range is between 4 and 28) |
| Generativity | Generativity is defined as the score of the revised generativity concern scale (GCS-R) (Range is between 0 and 28) |
| Inquisitiveness | Inquisitiveness is defined as the score by a subscale of the critical thinking disposition scale (Range is between 10 and 50) |
| Adult's positive response | Positive response is defined as the score on the positive effects subscale of the revised expected costs & benefits of the consultation scale (Range is between 8 and 40) |
| Adult's negative response | Negative response is defined as the score on the negative effects subscale of the revised expected costs & benefits of the consultation scale (Range is between 6 and 30) |
| Area | Area is the categorical variable of 0 and 1 where residential areas, non-urban areas and urban areas are coded as 0 and 1, respectively. |

concerning inquisitiveness. Third, we characterize inquisitiveness about positive responses to a child's questions along with other factors. We decide to rely on cross-sectional data following several previous studies which have argued for the effectiveness of cross-sectional data analyses in identifying correlation and causal relation among psychometric and sociodemographic variables, especially when the causal direction is somewhat obvious or intuitively straightforward [74, 75, 78]. Specifically, we use multivariate regression analyses to address three research questions: (1) How does children's inquisitiveness, along with generativity, affect happiness? (2) Does children's inquisitiveness play a role in generativity? (3) How does adults' manner of responding affect children's inquisitiveness?

To characterize research questions (1), (2) and (3), we perform regression analyses on the subjective happiness (SH), the revised generativity concern scale (GCS-R) and inquisitiveness are taken as dependent variables, respectively. When observations of these variables in the sample are considered to be non-normally distributed and/or skewed, the median regression is applied to analyze the determinants of dependent variables instead of parametric mean-based regressions, such as ordinary least squares (OLS) regression. We have run Shapiro-Wilk tests for three dependent variables of SH, GCS-R and inquisitiveness, as shown in Fig 2. The results reject the null hypothesis ($z = 3.955$, $P < 0.01$) for happiness, ($z = 5.871$, $P < 0.01$) for generativity and ($z = 5.741$, $P < 0.01$) for inquisitiveness. The tests confirm that the three variables, SH, GCS-R and inquisitiveness, are not normally distributed. The literature claims that median regressions are more appropriate than parametric mean-based ones, yielding robust estimations against the boundary values and/or outliers, especially when the dependent

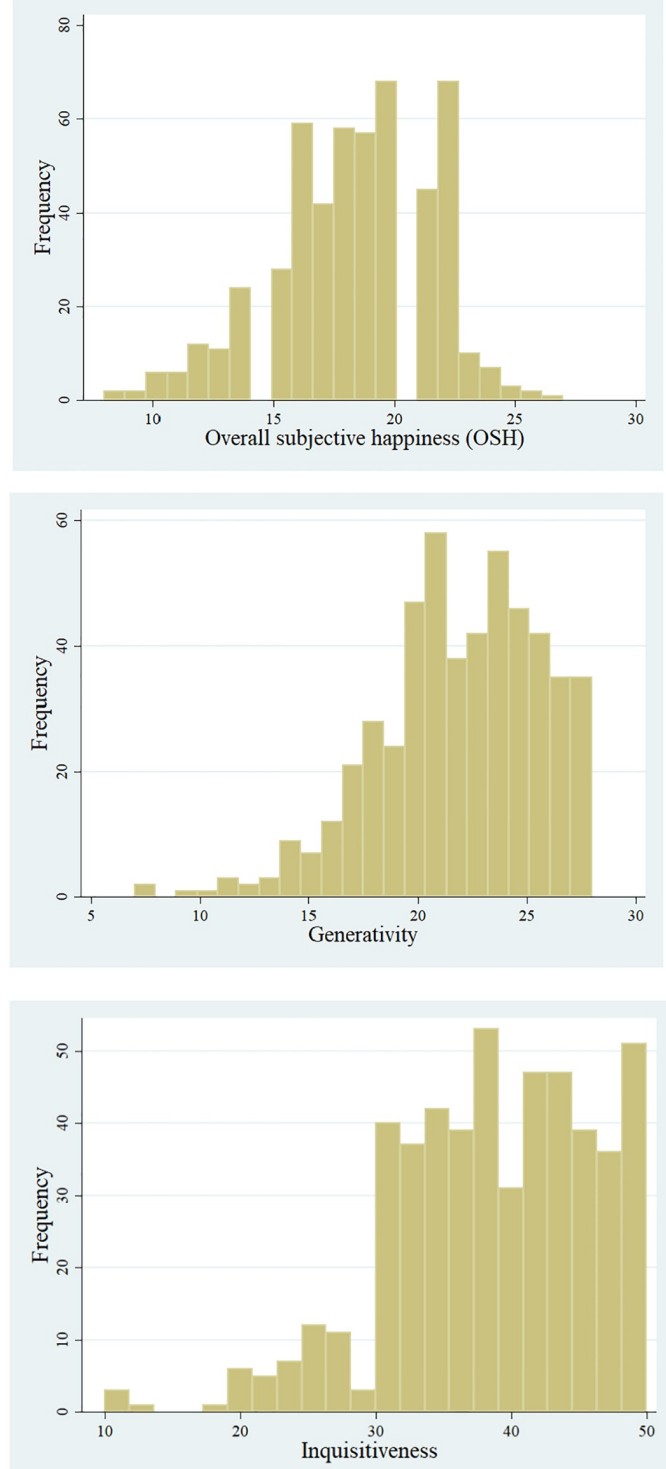

**Fig 2. Histograms of the dependent variables.** Histograms of the dependent variables, subjective happiness (SH), generativity (GCS-R) and inquisitiveness.

variable is bounded on a certain support range, non-normally distributed, and skewed [89, 90]. Therefore, we have employed the median regressions for happiness, generativity, and inquisitiveness with the specifications of Eqs 1 to 3, respectively.

Before the hypotheses test of regression models, it is evaluated by correlation matrix whether there is a very high level of correlation between the independent variables. Correlation analysis describes the relationship between variables A and B. If there is, the strength and direction of the relationship are analyzed; however, it is impossible to say which of the two variables is the cause or the result. The correlation matrix also presents a preliminary control on whether there is a multicollinearity problem (a very high correlation level) among the model's independent variables. The correlation matrix presented in Table 2 shows that the highest correlation in relationship with SH in adult' positive response ($r = 0.4910$; $p < 0.001$), with GCS-R in inquisitiveness ($r = 0.6575$; $p < 0.001$), with inquisitiveness in adult' positive response ($r = 0.5117$; $p < 0.001$). Table 2 indicates no other variables for which multicollinearity problems are observed in the regression models, with SH, GCS-R and inquisitiveness. Moreover, in selecting variables for the regression model, the survey design is created to test the hypotheses by taking into account perceptions of past facts, personality and emotions at the time of responding to the survey. Our study adjusts the direction of the dependent and independent variables by using, for example, the fact that they perceive a good response from an adult in the past, cognitive factors (i.e., GCS-R) and dispositions (i.e., inquisitiveness) that are assumed not to change quickly and SH, which strongly reflects emotions at the time of responding to the survey.

To this end, regression models are applied to characterize subjective happiness (SH), generativity (GCS-R) and inquisitiveness as dependent variables, respectively, in relation to other vital independent variables as described in Fig 1, enabling to identify of essential determinants. For empirically characterizing the SH of the participants $i$, the model is specified as

$$SH_i = \alpha_0 + \alpha_1 \cdot \text{inquisitiveness}_i + \alpha_2 \cdot \text{GCS} - R_i + \alpha_3 \cdot \text{adult's positive response}_i + \boldsymbol{\alpha}_4 \cdot \mathbf{x}'_i + \boldsymbol{\varepsilon}_i \quad (1)$$

where $SH_i$ stands for the participant $i$'s happiness. The coefficients, $\alpha_0, \alpha_1, \alpha_2, \alpha_3, \boldsymbol{\alpha}_4$, are

**Table 2. Correlation matrix.**

| Variables | 1 | 2 | 3 | 4 | 5 | 6 | 7 | 8 | 9 | 10 | 11 | 12 | 13 | 14 |
|---|---|---|---|---|---|---|---|---|---|---|---|---|---|---|
| 1. Gender (female) | 1.0000 | | | | | | | | | | | | | |
| 2. Age | 0.0080 | 1.0000 | | | | | | | | | | | | |
| 3. Household members | −0.0019 | −0.1528 | 1.0000 | | | | | | | | | | | |
| 4. Tutoring school days | −0.0830 | −0.0714 | 0.0084 | 1.0000 | | | | | | | | | | |
| 5. Joining sports clubs | −0.2305 | −0.0548 | 0.1374 | 0.0312 | 1.0000 | | | | | | | | | |
| 6. Having breakfast with family | −0.0545 | −0.0620 | −0.0605 | 0.0472 | 0.0607 | 1.0000 | | | | | | | | |
| 7. Having dinner with family | −0.0679 | 0.1000 | −0.1547 | −0.0642 | 0.0467 | 0.4161 | 1.0000 | | | | | | | |
| 8. Shopping for food with family | 0.0454 | −0.0305 | 0.0055 | 0.0765 | 0.0764 | 0.1656 | 0.1529 | 1.0000 | | | | | | |
| 9. Subjective happiness (SH) | 0.0398 | −0.0886 | −0.0713 | 0.0901 | 0.0814 | 0.1138 | 0.0421 | 0.1402 | 1.0000 | | | | | |
| 10. Generativity (GCS-R) | 0.1822 | −0.1862 | 0.0340 | 0.0711 | 0.1335 | 0.1519 | 0.0526 | 0.2120 | 0.3922 | 1.0000 | | | | |
| 11. Inquisitiveness | 0.0564 | −0.1722 | −0.0240 | 0.1211 | 0.1031 | 0.0914 | 0.0280 | 0.1875 | 0.4849 | 0.6575 | 1.0000 | | | |
| 12. Adult's positive response | −0.0064 | −0.0795 | −0.0080 | 0.1283 | 0.0696 | 0.1220 | −0.0155 | 0.1535 | 0.4910 | 0.4023 | 0.5117 | 1.0000 | | |
| 13. Adult's negative response | −0.0920 | −0.1298 | 0.0923 | 0.0250 | −0.0437 | −0.0054 | −0.0210 | 0.0341 | −0.2787 | −0.1404 | −0.2021 | −0.3313 | 1.0000 | |
| 14. Area (Urban) | −0.0082 | 0.1257 | −0.0753 | 0.2018 | −0.0945 | −0.0557 | −0.0257 | −0.0267 | −0.0370 | 0.0109 | 0.0513 | 0.0080 | −0.0395 | 1.0000 |

parameters to be estimated and $\varepsilon_i$ is an error term. In Eq 1, parameters $\alpha_1$ and $\alpha_2$ are of particular interest to statistically test question (1). For the generativity of the participant $i$, the model is

$$\text{GCS} - \text{R}_i = \beta_0 + \beta_1 \cdot \text{inquisitiveness}_i + \beta_2 \cdot \text{adult's positive response}_i + \boldsymbol{\beta}_3 \cdot \mathbf{x}_i' + \epsilon_i, \quad (2)$$

where $\mathbf{x}_i$ is a vector of sociodemographic variables, such as gender, age, household members and areas. The associated coefficients of $\beta_0, \beta_1, \beta_2, \boldsymbol{\beta}_3$ are the parameters to be estimated, and $\epsilon_i$ is an error term. In Eq 2, parameter $\beta_1$ is of particular interest to statistically examine question (2). For inquisitiveness of the participant $i$, the model is

$$\text{inquisitiveness}_i = \gamma_0 + \gamma_1 \cdot \text{adult's positive response}_i + \boldsymbol{\gamma}_2 \cdot \mathbf{x}_i' + \varepsilon_i \quad (3)$$

where adult's positive response$_i$ stands for the participant $i$'s positiveness to the adult's response. The coefficients, $\gamma_0, \gamma_1, \boldsymbol{\gamma}_2$, are parameters to be estimated and $\varepsilon_i$ is an error term. In Eq 3, parameters $\gamma_1$ is of particular interest to statistically test question (3).

To further confirm aforementioned regression results, we apply structural equation modeling (SEM) to check the types of relationships, i.e., "paths,": (1) inquisitiveness $\Rightarrow$ SH, (2) GCS-R $\Rightarrow$ SH, (3) adult's positive response $\Rightarrow$ SH, (4) adult's positive response $\Rightarrow$ inquisitiveness, (5) inquisitiveness $\Rightarrow$ GCS-R, (6) adult's positive response $\Rightarrow$ GCS-R. Especially, the existence paths are analyzed to establish whether or not "generativity (GCS-R)" is a mediator in the relationship between inquisitiveness and SH. Mediation is established as a concept to describe a relationship in which the first variable, $X$ (inquisitiveness), affects the second variable, $M$ (GCS-R) which then affects the third variable of the outcome, $Y$ (SH), where the second variable is called a "mediator" [91, 92]. The SEM is one of the effective methods that enables us to test the paths among the three variables together with the direct and indirect effects of inquisitiveness, following the procedures [93–95]. The SEM computes a beta weight as a standardized coefficient, ($\beta$), along with the associated statistical significance for each path. We can directly compare the magnitudes of standardized coefficients to estimate the relative strength of relationships. Standardization is necessary to compare direct and indirect effects among different sets of paths in the same model [96–98].

## Results

Table 3 presents the summary statistics for urban, non-urban and overall areas. The percentage of female participants and their mean age are almost similar in urban and non-urban areas, 46% and 47%, as well as 12.61 years and 12.21 years, respectively. Household members in urban and non-urban areas are 4.40 and 4.62, as shown in Table 3. The mean days of attendance at the tutorial school per week in urban and non-urban areas are 1.37 and 0.62, respectively. This result indicates the possibility of the urban household income being higher than that of non-urban areas because the tutoring school fee is expensive. The percentage of participants who join sports clubs in urban and non-urban areas is 39% and 50%, respectively. The mean time in which the participants had breakfast with the whole family per week in urban and non-urban areas is 2.87 and 3.24, respectively. The mean time in which the participants had dinner with the whole family per week in urban and non-urban areas is also 4.85 and 4.99, respectively. Co-eating food with family is thought to impact children's lifestyle, health and happiness positively [99–102]. Moreover, the mean time for family food shopping per week in urban and non-urban areas is also 1.56 and 1.75, respectively. Co-shopping–parents and children shopping together– is thought to positively influence children's social practice, consumer socialization and intergenerational interactions [103–106]. Table 3 shows the summary statistics of subjective happiness (SH) in urban, non-urban and overall areas. The Cronbach's alpha

**Table 3. Summary statistics of subject's sociodemographic information and major variables.**

| Variables | Urban areas | | | | | Non-urban areas | | | | | Overall areas | | | | |
|---|---|---|---|---|---|---|---|---|---|---|---|---|---|---|---|
| | M | Me | SD | Min | Max | M | Me | SD | Min | Max | M | Me | SD | Min | Max |
| Gender (female) | 0.46 | 0 | 0.50 | 0 | 1 | 0.47 | 0 | 0.50 | 0 | 1 | 0.46 | 0 | 0.50 | 0 | 1 |
| Age | 12.61 | 13 | 1.19 | 9 | 14 | 12.21 | 12 | 1.50 | 9 | 14 | 12.32 | 12 | 1.43 | 9 | 14 |
| Household members | 4.40 | 4 | 1.34 | 2 | 8 | 4.62 | 5 | 1.29 | 2 | 8 | 4.56 | 4 | 1.31 | 2 | 8 |
| Tutoring school days | 1.37 | 0 | 2.06 | 0 | 7 | 0.62 | 0 | 1.45 | 0 | 7 | 0.83 | 0 | 1.67 | 0 | 7 |
| Joining sports clubs | 0.39 | 0 | 0.49 | 0 | 1 | 0.50 | 0 | 0.50 | 0 | 1 | 0.47 | 0 | 0.50 | 0 | 1 |
| Having breakfast with family | 2.87 | 2 | 2.84 | 0 | 7 | 3.24 | 3 | 2.99 | 0 | 7 | 3.14 | 2 | 2.91 | 0 | 7 |
| Having dinner with family | 4.85 | 6 | 2.62 | 0 | 7 | 4.99 | 6 | 2.47 | 0 | 7 | 4.95 | 6 | 2.51 | 0 | 7 |
| Shopping for food with family | 1.56 | 0 | 2.48 | 0 | 8 | 1.75 | 1 | 2.38 | 0 | 8 | 1.70 | 1 | 2.11 | 0 | 8 |
| Subjective happiness (SH) | 18.09 | 18 | 3.1 | 9 | 25 | 18.35 | 19 | 3.30 | 8 | 27 | 18.28 | 19 | 3.22 | 8 | 27 |
| Generativity (GCS-R) | 22.21 | 23 | 3.96 | 7 | 28 | 22.12 | 22 | 3.91 | 7 | 28 | 21.15 | 22 | 3.92 | 7 | 28 |
| Inquisitiveness | 39.24 | 40 | 6.84 | 18 | 50 | 38.36 | 39 | 7.95 | 10 | 50 | 38.60 | 39 | 7.67 | 10 | 50 |
| Adult's positive response | 31.40 | 32.50 | 6.39 | 11 | 40 | 31.28 | 32 | 6.63 | 8 | 40 | 31.32 | 32 | 6.57 | 8 | 40 |
| Adult's negative response | 9.59 | 8 | 4.14 | 6 | 22 | 10.04 | 8 | 5.37 | 6 | 30 | 9.92 | 8 | 5.06 | 6 | 30 |
| Subjects | $n = 140$ | | | | | $n = 371$ | | | | | $n = 511$ | | | | |

SD stands for standard deviation.

we have computed for this scale is 0.93, illustrating that the overall subjective happiness (OSH) as subjective happiness scale (SHS) possesses acceptable internal consistency in our sample. The median scores of the OSH are 18 and 19 points in urban and non-urban areas, while the average scores of the OSH are 18.09 and 18.35 points, respectively. This finding suggests that SH between urban and non-urban participants is not different.

Table 3 shows the summary statistics of the participants' generativity (GCS-R) in urban, non-urban and overall areas. The Cronbach's alpha we have computed for this scale is 0.84, illustrating that the revised generativity concern scale possesses acceptable internal consistency in our sample. The median GCS-R scores are 23 and 22 points in urban and non-urban areas, respectively, while the average GCS-R scores are 22.21 and 22.12 points, respectively. This finding suggests that GCS-R between the urban and non-urban participants is similar; however, median GCS-R in the urban participants is slightly higher than that in the non-urban participants. Table 3 also shows the summary statistics of the participants' inquisitiveness in urban, non-urban and overall areas. The Cronbach's alpha we have computed for this scale is 0.90, illustrating that the inquisitiveness scale possesses acceptable internal consistency in our sample. The median scores of inquisitiveness are 40 and 39 points in urban and non-urban areas, while the average scores of inquisitiveness are 39.24 and 38.36 points, respectively. This finding suggests that inquisitiveness in the urban and non-urban participants is similar; however, generativity in the urban participants is slightly higher than that in the non-urban participants. We report the summary statistics of adult's positive responses in urban, non-urban and overall areas. In our sample, Cronbach's alpha for this subscale is 0.91. The median scores of inquisitiveness are 32.5 and 32.0 points in urban and non-urban areas, while the average scores of a positive response are 31.40 and 31.28 points, respectively. This finding suggests that positive responses in the urban and non-urban participants are not different.

To empirically characterize research question (1), we perform median regression in which SH is taken as a dependent variable, and inquisitiveness, GCS-R and adult's positive response are taken as independent variables along with other factors, as described in Eq 1. Table 4 reports the estimated coefficients ($\alpha_1, \alpha_2, \alpha_3, \boldsymbol{\alpha}_4$) and their respective standard errors in the

**Table 4. Estimation results of median regression on subjective happiness (SH).**

| Variables | Subjective happiness | | | |
|---|---|---|---|---|
| | **Model 1** | **Model 2** | **Model 3** | **Model 4** |
| Inquisitiveness | 0.208*** | 0.125*** | 0.107*** | 0.106*** |
| | (0.024) | (0.024) | (0.023) | (0.024) |
| Generativity (GCS-R) | 0.132*** | 0.028 | 0.015 | 0.026 |
| | (0.047) | (0.045) | (0.043) | (0.045) |
| Areas (base group = non- urban) | −0.698** | −0.707** | −0.834*** | −0.800*** |
| | (0.313) | (0.291) | (0.277) | (0.297) |
| Adult's positive response | | 0.202*** | 0.214*** | 0.210*** |
| | | (0.023) | (0.022) | (0.023) |
| Gender (base group = male) | | 0.040 | 0.046 | 0.066 |
| | | (0.263) | (0.250) | (0.271) |
| Age | | −0.040 | −0.081 | −0.063 |
| | | (0.092) | (0.089) | (0.093) |
| Household members | | | −0.162 | −0.201** |
| | | | (0.095) | (0.100) |
| Tutoring school days | | | | −0.009 |
| | | | | (0.079) |
| Joining sports clubs | | | | −0.132 |
| | | | | (0.270) |

***significant at 1 percent,

**significant at 5 percent,

*significant at 10 percent

independent variables on SH, along with statistical significance. Model 1 in Table 4 contains inquisitiveness, GCS-R and areas as independent variables. We gradually add adult's positive response, the gender dummy variable, age and other factors as independent variables in models 2 to 4, building upon model 1. First, we find that inquisitiveness is statistically significant with a positive sign at 1% in a robust manner, irrespective of the models. The estimated coefficients of inquisitiveness on the participants' SH ranged between 0.106 and 0.208 in models 1 to 4, implying that the participants may have an increase in the range of SH when one unit in their inquisitiveness rises.

Second, GCS-R has a positive effect on the participants' SH at 1% significance only in model 1. Previous studies with adults find the effect of generativity on subjective happiness statistically significant in a robust manner [11, 49, 70], while this study with children finds the effect statistically insignificant, except for model 1. Areas exhibit 1% and 5% statistical significance with a negative sign in models 1 to 4, implying that the urban participants tend to decrease their SH by −0.834 ∼ −0.698, as compared with the non-urban participants. We find that an adult's positive response is statistically significant with a positive sign at 1% in a robust manner in models 2 to 4. The estimated coefficients of adult's positive response to the participants' SH ranged between 0.202 and 0.214 in models 2 to 4, implying that the participants may have an increase in the range of SH when one unit in their experiences to receive "a positive response from adult" rises. The other independent variables, gender, age, household members, tutoring school days and joining sports clubs, are statistically insignificant, as shown in models 2 to 4 in Table 4. We confirm that the main results of inquisitiveness remain the same, irrespective of the various specifications of models other than models 1 to 4, such as the inclusion of age squared and/or interaction terms among the variables. On the other hand, while

previous studies with adults report that generativity is the most essential factor for subjective happiness (SH), this study with children shows very minimal expression of GCS-R compared to "adult's positive responses." Overall, inquisitiveness, adult's positive responses and residential area are identified as the main determinants of children's SH, and GCS-R is shown as insignificant in our models.

To empirically characterize research question (2), we perform the median regression in which the revised generativity concern scale (GCS-R) is taken as a dependent variable, and inquisitiveness is taken as an independent one along with other factors, as described in Eq 3. Table 5 reports the estimated coefficients ($\beta_1, \beta_2, \boldsymbol{\beta}_3$) and their respective standard errors of the independent variables on GCS-R, along with statistical significance. Model 1 of Table 5 contains inquisitiveness, gender and age as independent variables. Thereafter, we gradually add "adult's positive response" and other factors as independent variables in models 2 to 4, building upon model 1. First, we find that inquisitiveness is statistically significant with a positive sign at 1% in a robust manner, irrespective of the models. The estimated coefficients of inquisitiveness on the participants' GCS-R ranged between 0.303 and 0.348 in models 1 to 4, implying an increase in the participants' GCS-R range when one unit in their inquisitiveness rises.

Second, gender positively affects GCS-R at 1% significance in models 1 and 4. The estimated coefficients of gender in models 1 to 4 suggest that the participants may have an increase in their GCS-R range between 1.064 and 1.123 when they are females. In models 1 to 4, the participants may have a decrease in their GCS-R range by $-0.304 \sim -0.212$ at 1% and 5% significance when they age by one year. Moreover, "adult's positive response" exhibits 1% statistical significance with a positive sign in models 2 to 4, implying that the participants who

**Table 5. Estimation results of median regression on the revised generativity concern scale (GCS-R).**

| Variables | Generativity | | | |
|---|---|---|---|---|
| | Model 1 | Model 2 | Model 3 | Model 4 |
| Inquisitiveness | 0.348*** | 0.313*** | 0.304*** | 0.303*** |
| | (0.019) | (0.023) | (0.021) | (0.023) |
| Gender (base group = male) | 1.087*** | 1.065*** | 1.064*** | 1.123*** |
| | (0.280) | (0.296) | (0.279) | (0.305) |
| Age | −0.304*** | −0.240** | −0.239** | −0.212** |
| | (0.099) | (0.104) | (0.109) | (0.107) |
| Adult's positive response | | 0.083*** | 0.083*** | 0.083*** |
| | | (0.026) | (0.025) | (0.026) |
| Having breakfast with the whole family | | | 0.017 | 0.084 |
| | | | (0.053) | (0.056) |
| Having dinner with the whole family | | | 0.085 | 0.039 |
| | | | (0.062) | (0.066) |
| Household members | | | | 0.059 |
| | | | | (0.116) |
| Tutoring school days | | | | -0.098 |
| | | | | (0.091) |
| Joining sports clubs | | | | 0.403 |
| | | | | (0.308) |
| Area (base group = non-urban) | | | | -0.116 |
| | | | | (0.342) |

***significant at 1 percent,

**significant at 5 percent,

*significant at 10 percent

perceive an adult's positive response in the past enhance their GCS-R by 0.083, as compared with other participants. The other independent variables, such as "having a breakfast with the whole family," "having a dinner with the whole family," "household members," "tutoring school days," "joining sports clubs" and "residential areas," are identified to be statistically insignificant, as shown in models 3 and 4 in Table 5. We confirm that the main results qualitatively remain the same, irrespective of the various specifications of models other than models 1 to 4, such as interaction terms among the variables. Overall, inquisitiveness, gender, age and adult's positive response are the main determinants having statistical and practical significance to the likelihood of a participant's increasing GCS-R.

To empirically characterize research question (3), we perform the median regression with inquisitiveness is taken as a dependent variable and positive response is taken as an independent one along with other factors, as described in Eq 3. Table 6 reports the estimated coefficients ($\gamma_1$, $\gamma_2$) and their respective standard errors of the independent variables on inquisitiveness, along with statistical significance. Model 1 of Table 6 contains positive response and age as independent variables. Next, we gradually add "family food shopping," "joining sports clubs" and other factors as independent variables in models 2 to 4, building upon model 1. First, we find that the positive response is statistically significant with a positive sign at 1% in a robust manner, irrespective of the models. The estimated coefficients of positive response on the participants' inquisitiveness range between 0.657 and 0.717 in models 1 to 4, implying the likelihood of an increase in the participants' inquisitiveness when one unit in their positive response level rises.

Second, age negatively affects inquisitiveness at 1% significance in models 1 and 4. The estimated coefficients of age in models 1 to 4 suggest that the participants may have a decrease in

**Table 6. Estimation results of median regression on child's inquisitiveness.**

| Variables | Inquisitiveness | | | |
|---|---|---|---|---|
| | Model 1 | Model 2 | Model 3 | Model 4 |
| Posotive response | 0.717*** | 0.673*** | 0.680*** | 0.657*** |
| | (0.046) | (0.056) | (0.057) | (0.056) |
| Age | -0.770*** | -0.901*** | -0.904*** | -0.957*** |
| | (0.210) | (0.242) | (0.246) | (0.244) |
| Family food shopping | | 0.383** | 0.324** | 0.377** |
| | | (0.150) | (0.151) | (0.146) |
| Joining sports clubs | | 0.914 | 1.099 | 1.536** |
| | | (0.686) | (0.718) | (0.709) |
| Negative response | | -0.074 | -0.059 | -0.041 |
| | | (0.073) | (0.074) | (0.073) |
| Gender (base group = male) | | | 0.885 | 1.273* |
| | | | (0.712) | (0.702) |
| Household members | | | -0.115 | -0.323 |
| | | | (0.269) | (0.265) |
| Tutoring school days | | | | 0.135 |
| | | | | (0.210) |
| Area (base group = non-urban) | | | | -1.251 |
| | | | | (0.783) |

***significant at 1 percent,

**significant at 5 percent,

*significant at 10 percent

the range of their inquisitiveness between -0.770 and -0.957 when they age by one year. The negative value indicated by age is consistent with several studies that report a negative linear transition of curiosity, a concept similar to inquisitiveness, once young children begin formal schooling [107–109]."Family food shopping" also exhibits 5% statistical significance with a positive sign in models 2 to 4, implying that the participants who go grocery shopping with their parents enhance their inquisitiveness $0.324 \sim 0.383$, as compared with other participants. Several independent variables, such as "joining sports clubs," "negative response," gender and areas show the possibility as minor determinants of the participants' inquisitiveness. The other independent variables, such as household members and tutoring school days, are statistically insignificant, as shown in models 2 to 4 in Table 6. We confirm that the main results qualitatively remain the same, irrespective of the various specifications of models other than models 1 to 4, such as interaction terms among the variables. Overall, positive response, age and "family food shopping" are confirmed as the main determinants having statistical and practical significance to the likelihood of the participants increasing their inquisitiveness.

The multivariable regression analysis, for example, Eq 1, can reveal the impact of solely the explanatory variable on the dependent variable of subjective happiness (SH), with the effect (i.e., indirect effect) of other explanatory variables being irrelevant (i.e., Frisch-Waugh theorem) [90]. Therefore, we employ structural equation modeling (SEM) to test the hypotheses about mediating effects between variables in Eq 1, as graphically conceptualized in Fig 3. Our regression analyses are a simple form of partial regression coefficients, but the SEM provides more advanced capabilities for complex modeling relationships between variables. First, we analyze the three direct effects of inquisitiveness on SH (path *A* in Fig 3), of GCS-R on SH (path *B* in Fig 3) and of adult's positive response on SH (path *C* in Fig 3) by SEM standardized analysis. The results show the existence of path *A* with ($\beta = 0.317$, $p < 0.000$), that of path *B* ($\beta = 0.092$, $p = 0.059$) and that of path *C* ($\beta = 0.329$, $p < 0.000$), meaning that these three variables appear to have direct effects on SH, respectively.

Second, we analyze the direct effects of inquisitiveness on GCS-R (path *D* in Fig 3) and an indirect effect of inquisitiveness on SH through GCS-R (path $\hat{A}$ in Fig 3). The SEM analysis demonstrates the significance of path *D* ($\beta = 0.612$, $p < 0.000$) as well as path $\hat{A}$ ($\beta = 0.056$, $p < 0.000$). Based on these results, we confirm that the indirect path $\hat{A}$ from inquisitiveness to SH plays a small but positive role through a mediator of GCS-R. Third, we analyze the direct effects of adult's positive response on GCS-R (path E in Fig 3) and of adult's positive response on inquisitiveness (path F in Fig 3) by SEM standardized analysis. The SEM analysis demonstrates the significance of path E ($\beta = 0.402$, $p < 0.000$) as well as path F ($\beta = 0.512$, $p < 0.000$). Furthermore, we analyze an indirect effect of adult's positive response on GCS-R through inquisitiveness (path $\hat{E}$ in Fig 3). The results show the existence of path $\hat{F}$ ($\beta = 0.313$, $p < 0.000$), meaning that adult's positive response appear to have an indirect effect on GCS-R through inquisitiveness.

Because our results infer the importance of inquisitiveness and adults' positive response over GCS-R for SH in children, we perform the SEM analysis of these three variables in Fig 4. First, we analyze the two direct effects of inquisitiveness on SH (path *A* in Fig 4) and of adult's positive response on SH (path *C* in Fig 4) by SEM standardized analysis. The results successfully show the existence of path *A* with ($\beta = 0.317$, $p < 0.000$) and that of path *C* ($\beta = 0.329$, $p < 0.000$), meaning that both inquisitiveness and adult's positive response appear to have direct effects on SH. Second, we analyze the direct effects of an adult's positive response on inquisitiveness (path E in Fig 4) and an indirect effect of an adult's positive response on SH through inquisitiveness (path $\hat{C}$ in Fig 4). The SEM analysis demonstrates the significance of path E ($\beta = 0.512$, $p < 0.000$) as well as path $\hat{C}$ ($\beta = 0.662$, $p < 0.000$). Based on these results,

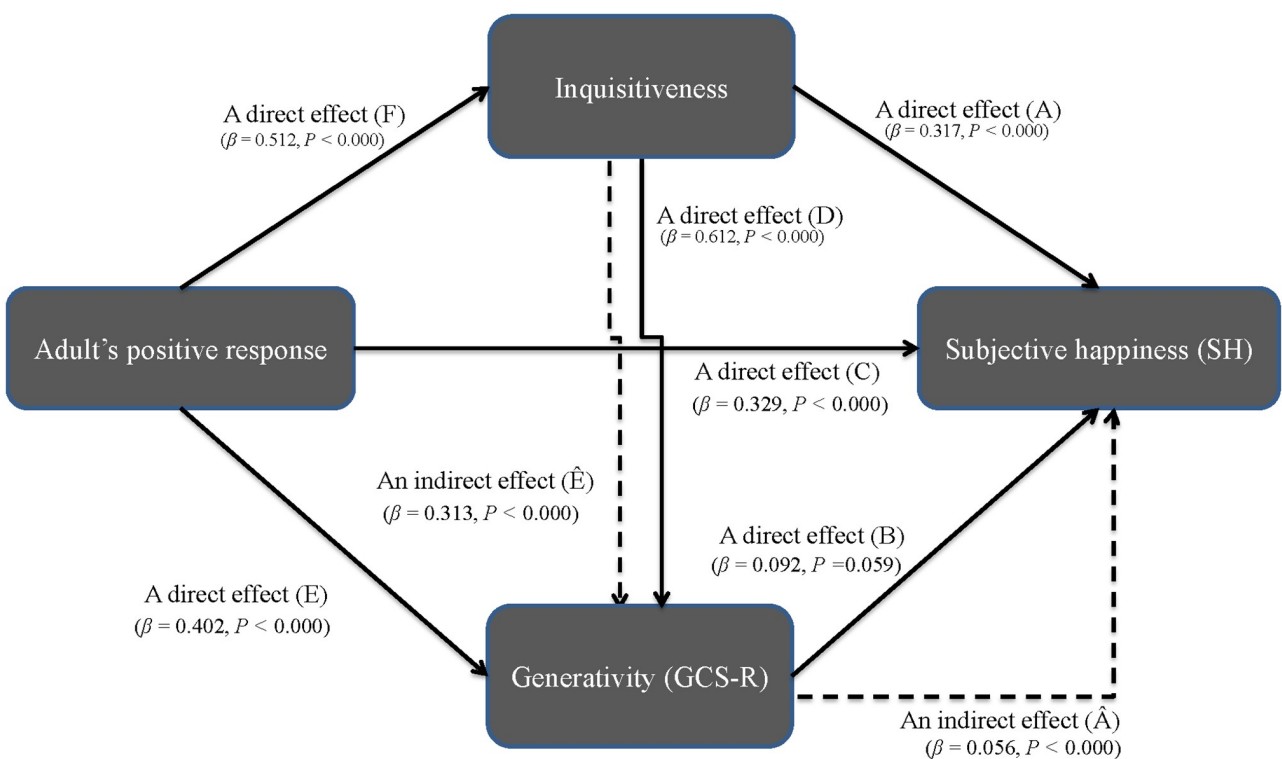

**Fig 3. The mediating effects (1).** The mediating effects among adult's positive response, inquisitiveness, GCS-R and SH.

we confirm that the indirect path $\hat{C}$ from an adult's positive response to SH plays a positive role through a mediator of inquisitiveness. Overall, the SEM analysis reconfirms the regression analysis results that inquisitiveness, GCS-R and adult's positive response, directly and indirectly, affect SH.

We now summarize the answers to the three research questions at the end of the introduction. As described in our conceptual framework of Fig 1, it is well known that cognitive, non-cognitive and sociodemographic factors mainly characterize happiness. The first question is,

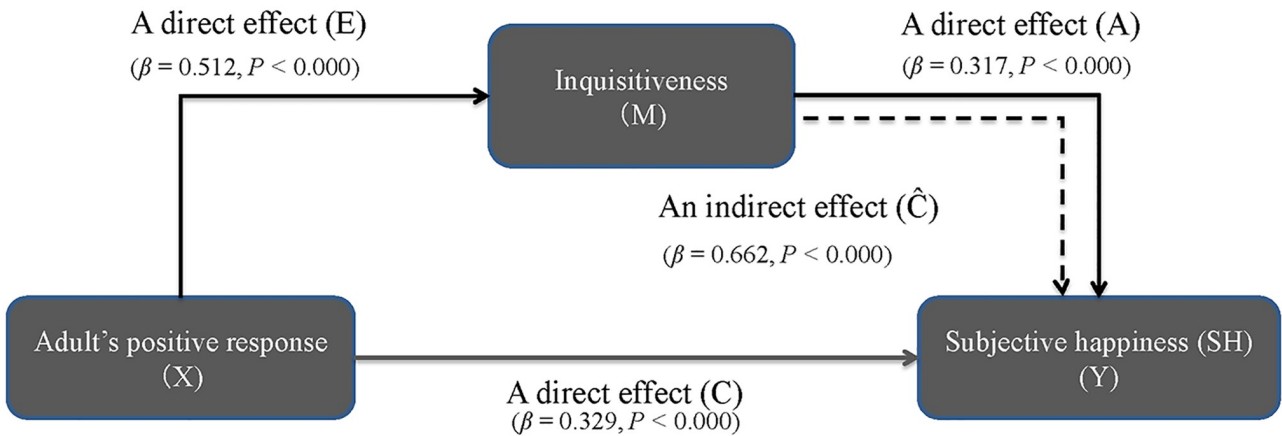

**Fig 4. The mediating effects (2).** The mediating effects among adult's positive response, inquisitiveness and SH.

"How does children's inquisitiveness, along with generativity, affect happiness?" We answer that generativity ($\alpha_1$), inquisitiveness ($\alpha_2$) and adult's positive response ($\alpha_3$), directly and indirectly, affect happiness in childhood, demonstrating the importance of generativity and inquisitiveness for subjective happiness (SH) in Fig 1. The second question is, "Does children's inquisitiveness play a role in generativity?" We answer that inquisitiveness ($\beta_1$) is meaningful to ascertain whether children possess the signs of generativity, along with adult's positive response ($\beta_2$) in Fig 1. The third question is, "How does adults' response manner affect children's inquisitiveness?" We answer that adult's positive response ($\gamma_1$) is the crucial determinant to ascertain whether children possess high inquisitiveness in Fig 1. Previous studies indicate the importance of generativity for an adult's subjective happiness [11, 49, 70]. However, our regression and SEM analyses show that the magnitude and statistical significance of inquisitiveness and adults' positive responses are more important for subjective happiness in their childhood than generativity.

## Discussions

We should mention the inverse correlation between the children's age and their inquisitiveness, shown in Table 6. The negative correlation, as revealed by the regression analysis in our study, is consistent with several previous studies reporting a linear negative transition between children's age and curiosity [107–109]. Many studies report a general decline in questioning throughout children and adolescence [107, 108, 110–112]. For instance, Tizard and Hughes [107] find that children reduce their questions from twenty-six an hour at home with their mothers to two per hour upon entering the formal education system. While we understand the importance of distinguishing curiosity from inquisitiveness, both share concepts that help us predict the nexus between age and inquisitiveness. Other studies cast doubt upon the veracity of the purported linear transition in age and curiosity, advocating instead for the pivotal role of educators and environmental factors in nurturing children's inquisitive disposition [113, 114]. Regrettably, our dataset remains insufficient to reveal the mechanism of the negative correlation between children's age and their inquisitiveness. Nevertheless, our study proffers empirical evidence that suggests the hypothesis that constructive adult responses to children's inquiries may positively correlate with the young ones' inquisitiveness.

Hirose and Kotani [70] and Hirose et al. [11] have pointed out that inquisitiveness positively correlates with how people enhance their generative concern and happiness through questionnaire surveys with Japanese and Palauan adults, respectively. According to Hirose and Kotani [70] and Hirose et al. [11], inquisitiveness, directly and indirectly, affects happiness, but the most critical factor is generativity. Our study also demonstrates that children's degree of inquisitiveness is positively correlated with their generativity and happiness. Some studies have pointed out that inquisitiveness is stable as a part of one's critical thinking disposition [68, 112, 115]. Conversely, other studies have pointed out that inquisitiveness can be acquired and further enhanced by learning [6, 7, 58, 116].

Inquisitiveness and adults' positive responses are identified as the main determinants of children's subjective happiness, and our regression models show generativity as insignificant. Generativity is the characteristic that helps and leads the next generation to grow [8–11]. Childhood is also important as a preparatory process for generativity, but for children, it is more critical to arouse and satisfy inquisitiveness through good relationships with adults than with the problems of future generations. Although limited by data from self-reported answers at one point in time, our study demonstrates that children's inquisitiveness is strongly and positively associated with the perception that they received an encouraging response to their inquisitiveness from an adult and with signs of their generative concern. Overall, the results of

this study suggest that learning together in intergenerational relationships between adults and children is essential for people's subjective happiness and should be considered for development as an educational program for families, communities and schools.

We state some limitations in our research and suggest directions to address them in future research through intergenerational relations. Based on the above discussions, we can consider that inquisitiveness can increase through education, training and daily experiences. Our next question is whether adults with relevant childhood experiences of question-answer exchanges among generations improve generativity and respond better in their adulthood to the younger ones' questions than otherwise. This study does not address the long-term observation that the participants who experienced having received good adult responses in childhood can improve their generativity in adulthood. Some studies have argued that collecting and examining longitudinal data is more desirable than dealing with cross-sectional data to confirm our findings' robustness and consistency and analyze our new questions [117–119]. Therefore, future studies should conduct experimental projects to collect longitudinal data and examine causality among inquisitiveness, generativity and happiness. These limitations notwithstanding, we believe that this research is an essential first step toward understanding the importance of children's inquisitiveness, generativity and happiness. We hope that further studies will identify how to pass the "baton of critical thinking" to successive generations to contribute to a sustainable human society.

## Conclusions

This paper addresses how children's inquisitiveness influences their generative concern and happiness through intergenerational relationships. We claim that inquisitiveness is essential for children to enhance their happiness, hypothesizing that children who receive a positive response from adults tend to be inquisitive and express the signs of generativity. To empirically examine the hypothesis, we statistically characterize inquisitiveness in relation to adult-child interaction, generativity (offering concern and care for people & the natural environment), and happiness with the data from a survey of 511 Japanese children between 9 and 14 years of age by applying the revised generativity concern scale (GCS-R). The results show that inquisitiveness correlates with generativity and happiness, primarily that a positive response by adults to children's inquiries promotes their inquisitiveness through adult-child interactions. Our analysis shows that children's inquisitiveness, encouraged by adults' positive responses, is more significant in happiness than the generativity concern during childhood. Overall, the results suggest that adults responding positively to children's questions is essential for promoting inquisitiveness and increasing happiness.

## Supporting information

**S1 File. It contains all the necessary data to replicate the statistical and regression results presented in this paper.**
(XLSX)

**S1 Checklist.** *PLOS ONE* **clinical studies checklist.**
(DOCX)

## Acknowledgments

The author is grateful to anonymous referees for their valuable comments and feedback. The author also would like to thank Koji Kotani, Moinul Islam, Koji Funakoshi, Masahiro Ogawa and Toshihisa Kariya for their helpful advice and comments.

## Author Contributions

**Conceptualization:** Junichi Hirose.

**Data curation:** Junichi Hirose.

**Formal analysis:** Junichi Hirose.

**Funding acquisition:** Junichi Hirose.

**Investigation:** Junichi Hirose.

**Methodology:** Junichi Hirose.

**Project administration:** Junichi Hirose.

**Resources:** Junichi Hirose.

**Software:** Junichi Hirose.

**Supervision:** Junichi Hirose.

**Validation:** Junichi Hirose.

**Visualization:** Junichi Hirose.

**Writing – original draft:** Junichi Hirose.

**Writing – review & editing:** Junichi Hirose.

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
