## [Decision Letter · Decision Letter 0]

31 Aug 2023

PONE-D-23-14243How do question-answer exchanges among generations matter for children’s happiness?PLOS ONE

Dear Dr. Hirose,

Thank you for submitting your manuscript to PLOS ONE. After careful consideration, we feel that it has merit but does not fully meet PLOS ONE’s publication criteria as it currently stands. Therefore, we invite you to submit a revised version of the manuscript that addresses the points raised during the review process.

ACADEMIC EDITOR:The reviewers have highlighted areas for improvement to enhance the current version of the manuscript:

Regarding Reviewer #1, a clearer rationale is needed for choosing quantile regression over OLS. Additionally, in the data analysis section (lines 230 to 235), a more detailed explanation about the interaction among key variables is necessary, possibly employing a correlation matrix to elucidate this interaction. It is also suggested to explore the relationship between generativity and SH within different groups determined by the confounding variables.

Reviewer #2 pointed out that the research questions addressed in the manuscript are both novel and important. However, they noted that the rationale for these questions in the Introduction is at times unclear. It is recommended to avoid distracting statements and to define key terms, such as "generative concern," clearly right from the beginning of the Introduction. Furthermore, providing a high-level overview of the methodological and analytical approach towards the end of the Introduction would be beneficial. The reviewer also provided several specific suggestions to enhance the manuscript's presentation, from term definitions to section organization.

Therefore, the reviewers' recommendations include clarifying key terms, improving the rationale for the research questions, providing a more detailed description of the methodology and analytical approach, streamlining and organizing the manuscript sections more effectively, and addressing minor errors to ensure the manuscript is clear, cohesive, and rigorous in its presentation.

We look forward to receiving your revised manuscript.

Kind regards,

Ricardo Limongi

Academic Editor

PLOS ONE

Journal Requirements:

"YES-The funders provided the author with a portion of the research funds as well as cooperation in obtaining the approval of the local school boards to recruit subjects."

6. We note that Figure 1 in your submission contain map/satellite images which may be copyrighted. All PLOS content is published under the Creative Commons Attribution License (CC BY 4.0), which means that the manuscript, images, and Supporting Information files will be freely available online, and any third party is permitted to access, download, copy, distribute, and use these materials in any way, even commercially, with proper attribution. For these reasons, we cannot publish previously copyrighted maps or satellite images created using proprietary data, such as Google software (Google Maps, Street View, and Earth). For more information, see our copyright guidelines: http://journals.plos.org/plosone/s/licenses-and-copyright.

Reviewers' comments:

Reviewer's Responses to Questions

**Comments to the Author**

1. Is the manuscript technically sound, and do the data support the conclusions?

Reviewer #1: Yes

Reviewer #2: Partly

2. Has the statistical analysis been performed appropriately and rigorously? 

Reviewer #1: Yes

Reviewer #2: I Don't Know

3. Have the authors made all data underlying the findings in their manuscript fully available?

Reviewer #1: Yes

Reviewer #2: Yes

4. Is the manuscript presented in an intelligible fashion and written in standard English?

Reviewer #1: Yes

Reviewer #2: Yes

5. Review Comments to the Author

Reviewer #1: 1) Clear justification for using the quantile regression against OLS

2) In data analysis part lines 230 to 235: the confounding interaction among key variables needs more explanation. I think a correlation matrix might assist. Also, examining the relationship between generativity and SH according to different groups or levels of the confounding variables.

Reviewer #2: The research questions addressed in this manuscript are both novel and important. However, the rationale for the questions in the Introduction is sometimes difficult to follow; for example, there are some distracting claims in the introduction (e.g., the statement about Maslow, noted below), and key terms (e.g., "generative concern") need to be explained more clearly and directly early on in the Introduction. A general (high-level) orientation to the methodological and analysis approach for the study in the final part of the Introduction would also be valuable. Elaboration on these and additional general points, and more specific comments, are provided below.

(1) Introduction: The meaning of the phrase "generative concern" is not clear early on in the Introduction. Please explicitly state what is meant by this phrase early in the Introduction (e.g., in the first paragraph). For example, on page 6 it is characterized as "offering care for people and (the) natural environment." Providing a few examples/paraphrases of the actual questions given to the young age-group tested could also be helpful.

(2) Introduction: It is a considerable mischaracterization of Maslow's theory to suggest that "people are happy as they become wealthy." Please rephrase.

(3) The subsection on "Participants" is an admixture of study overview statements, participants, and procedure.

(a) It should be greatly condensed, using more direct and simple statements, that specifically describe the participants.

(b) It could be helpful for some of the demographic information that is presented in the first section of the results (e.g., percentage of females and mean age in urban and non-urban areas) to be instead provided here in the Participants subsection.

(c) Please indicate how many participants were invited to take part vs. how many actually participated.

(d) Some of the text in the Participants subsection could be better presented earlier, as a study overview or similar (e.g., the broad characterization of the measures and the reference to Figure 1). Perhaps provide such an overview in a paragraph or two before the Method section.

(e) Some of the text in the Participants subsection, such as sending invitation letters and obtaining written consent should be in a separate Procedure subsection.

(f) The procedure subsection should also provide additional details of how the study was actually conducted, such as how long it took for the children to complete the questionnaires, the order in which the questionnaires were completed, when the questionnaires were administered (e.g., in the children's classrooms, during class-time or before or after school) and how they were administered (e.g., in computerized or paper-and-pencil format).

(4) Discussion -- The first paragraph of the discussion, which has multiple specific references to Figure 2, seems more appropriately placed as a concluding paragraph of the Results.

(5) Discussion -- The sentence, "Our study demonstrates that children who receive suitable responses from adults tend to increase their inquisitiveness and express the signs of generativity" is an over-statement of what the study can show (given it is an entirely cross-sectional study, using only self-reported outcomes). What the study shows is that (at one point in time) children's self-reported inquisitiveness is strongly positively associated with their (self-reported) perception of receiving encouraging responses for their inquisitiveness from adults, and also is positively associated with children's (self-reported) generativity concerns.

(6) What is to be made of the negative relation between children's inquisitiveness and children's age (e.g., page 17 and Table 5)? There should be some explicit discussion of this outcome.

Minor Comments

-- p. 2, a manuscript submitted to a scientific journal would not usually include a "contents" page (a contents page is generally more expected/appropriate for a book or a thesis)

-- p. 5, words missing/fragmented sentence: "Therefore, inquisitiveness is hypothesized to be a crucial determinant of happiness and generativity, even among children, and empirically examine their cognitive, noncognitive and sociodemographic factors using a single analytical framework." Suggest separating these key ideas into two sentences, along the lines: "Therefore, inquisitiveness is hypothesized to be a crucial determinant of happiness and generativity, even among children. The aim of this study is to empirically examine the cognitive, noncognitive and sociodemographic factors of inquisitiveness, generativity and happiness in children (9 to 14 years of age) using a single analytical framework."

-- p. 7, the subsection heading should read "Measures"

-- p. 9 - p. 10, the word "theoretical" in "theoretical range" (several places) is confusing; suggest rephrasing as "the maximum possible range"

-- p. 10: "These subscales are reliable for influencing people's behaviors and attitudes in many vital contexts" -- suggest rephrasing to read something like, "These subscales have been found to reliably assess people's behaviors and attitudes in many important contexts"

-- p. 11 - p. 12 -- rephrase "disturbance term" (several places) as "error term"

-- p. 12 -- the rationale for performing median regression analyses might be better placed earlier in the results section.

-- p. 15 - p. 16: important typo: the text here should read "Overall.... confirmed to be the main determinants of the participants' subjective happiness (SH)" (not of the participants' generativity")

-- p. 18 -- the relevant Figure number for path C is not indicated

-- p. 18 -- the subsection heading should read "Discussion"

-- p. 18 -- suggest rephrasing to read "the three research questions posed"

-- p. 19 -- the text, "children’s inquisitiveness statistically affects the sign of generativity and happiness" is unclear/difficult to parse; suggest something like "children's degree of inquisitiveness is significantly positively correlated with their generativity and happiness"

-- p. 20 -- the reference to "panel data" is unclear; suggest "longitudinal data" or similar

-- Figure 2 -- the text in the Figure should read "sports clubs" (not ports clubs)

6. PLOS authors have the option to publish the peer review history of their article (what does this mean?). If published, this will include your full peer review and any attached files.

Reviewer #1: **Yes: **Ali Abdallah

Reviewer #2: No

---

## [Author Response · Author response to Decision Letter 0]

14 Oct 2023

To Reviewer 1

I greatly appreciate your very constructive comments. Your comments have made me ponder over the conceptual framework. As a result, my paper has been improved. Please see the separate document for more details. Thank you very much.

To Reviewer 2

I greatly appreciate your very kind advice and suggestions. Your comments have been a great learning experience for me. As a result, my paper has been improved. Please see the separate document for more details. Thank you very much.

---

## [Decision Letter · Decision Letter 1]

26 Apr 2024

How do question-answer exchanges among generations matter for children’s happiness?

PONE-D-23-14243R1

Dear Dr. Hirose,

We’re pleased to inform you that your manuscript has been judged scientifically suitable for publication and will be formally accepted for publication once it meets all outstanding technical requirements.

Kind regards,

Ricardo Limongi

Academic Editor

PLOS ONE

Additional Editor Comments (optional):

Reviewers' comments:

Reviewer's Responses to Questions

**Comments to the Author**

1. If the authors have adequately addressed your comments raised in a previous round of review and you feel that this manuscript is now acceptable for publication, you may indicate that here to bypass the “Comments to the Author” section, enter your conflict of interest statement in the “Confidential to Editor” section, and submit your "Accept" recommendation.

Reviewer #1: All comments have been addressed

Reviewer #3: All comments have been addressed

2. Is the manuscript technically sound, and do the data support the conclusions?

Reviewer #1: Yes

Reviewer #3: Yes

3. Has the statistical analysis been performed appropriately and rigorously? 

Reviewer #1: Yes

Reviewer #3: Yes

4. Have the authors made all data underlying the findings in their manuscript fully available?

Reviewer #1: Yes

Reviewer #3: Yes

5. Is the manuscript presented in an intelligible fashion and written in standard English?

Reviewer #1: Yes

Reviewer #3: Yes

6. Review Comments to the Author

Reviewer #1: Here are two minor recommendation might add more insights and clearance to the article:

1- in line 242-243 you mentioned in the third research question 3) How does adult's manner of responding ....

please make sure to highlight the answer about the relationship between adult's response and children's inquisitiveness in abstract and conclusions parts.

2- For the word appropriately in the last two lines in the abstract and the conclusion, it will be more insightful if you replace it by more explanatory word such as positive or any other evidenced adjective.

Reviewer #3: The new version is clear enough about the contributions, and I think you should accept the article that will help future researchers.

7. PLOS authors have the option to publish the peer review history of their article (what does this mean?). If published, this will include your full peer review and any attached files.

Reviewer #1: **Yes: **Ali Abdallah

Reviewer #3: No

---

## [Editor Report · Acceptance letter]

27 May 2024

PONE-D-23-14243R1 

PLOS ONE

Dear Dr. Hirose, 

I'm pleased to inform you that your manuscript has been deemed suitable for publication in PLOS ONE. Congratulations! Your manuscript is now being handed over to our production team.

Kind regards, 

on behalf of

Professor Ricardo Limongi 

Academic Editor

PLOS ONE